# Relaxation of social distancing restrictions: Model estimated impact on COVID-19 epidemic in Manitoba, Canada

**Leigh Anne Shafer** [1,2]*, **Marcello Nesca**[2,3], **Robert Balshaw**[3]

**1** Department of Internal Medicine, University of Manitoba, Winnipeg, Manitoba, Canada, **2** Department of Community Health Sciences, University of Manitoba, Winnipeg, Manitoba, Canada, **3** George and Fay Yee Centre for Healthcare Innovation, University of Manitoba, Winnipeg, Manitoba, Canada

* shafer@umanitoba.ca

**Data Availability Statement:** All relevant data are within the manuscript and its Supporting Information files.

**Funding:** The authors have no financial disclosures to report.

## Abstract

### Objectives

The unprecedented worldwide social distancing response to COVID-19 resulted in a quick reversal of escalating case numbers. Recently, local governments globally have begun to relax social distancing regulations. Using the situation in Manitoba, Canada as an example, we estimated the impact that social distancing relaxation may have on the pandemic.

### Methods

We fit a mathematical model to empirically estimated numbers of people infected, recovered, and died from COVID-19 in Manitoba. We then explored the impact of social distancing relaxation on: (a) time until near elimination of COVID-19 (< one case per million), (b) time until peak prevalence, (c) proportion of the population infected within one year, (d) peak prevalence, and (e) deaths within one year.

### Results

Assuming a closed population, near elimination of COVID-19 in Manitoba could have been achieved in 4–6 months (by July or August) if there were no relaxation of social distancing. Relaxing to 15% of pre-COVID effective contacts may extend the local epidemic for more than two years (median 2.1). Relaxation to 50% of pre-COVID effective contacts may result in a peak prevalence of 31–38% of the population, within 3–4 months of initial relaxation.

### Conclusion

Slight relaxation of social distancing may immensely impact the pandemic duration and expected peak prevalence. Only holding the course with respect to social distancing may have resulted in near elimination before Fall of 2020; relaxing social distancing to 15% of pre-COVID-19 contacts will flatten the epidemic curve but greatly extend the duration of the pandemic.

**Competing interests:** The authors have declared that no competing interests exist.

## Introduction

In January 2020, a coronavirus called SARS-CoV-2 was identified [1] as the causative agent of the new coronavirus disease 2019 (COVID-19) outbreak in Wuhan, China, in December 2019 [1]. COVID-19 has since spread rapidly to every country in the world, thus has been declared a pandemic [2,3]. As of May 13 2020, nine days after the first phase of social distancing relaxation began in Manitoba Canada, an estimated 4.17 million people worldwide were estimated to have contracted COVID-19, with over 288,000 deaths [4]. The unprecedented worldwide response in the form of social distancing [5–7] resulted in a quick reversal of the rising numbers of COVID-19 cases in many geographic areas [8–10]. Beginning in May, local and regional governments worldwide began to relax social distancing regulations [11,12]. Since then, infections have risen. By July 27, 2020, infections had risen to an estimated 16.11 million people worldwide and 646,000 deaths [13]. Using the situation in Manitoba, Canada as an example, we aimed to estimate the impact that continued relaxation of social distancing regulations may have on local COVID-19 epidemics throughout the world, in the absence of vaccination.

Throughout this paper, we sometimes refer to the 'epidemic' to distinguish the situation in Manitoba from the global COVID-19 pandemic. For reference, in the 2016 census, 59% of the Manitoban population was concentrated in the two largest urban areas comprising 705,000 (55%) and 48,900 (4%). Population densities reflect this urban concentration, with corresponding densities of 1,519 and 631 per square kilometre. Overall population density is 2.3 per square kilometre [14,15]. The public health emergency was declared in Manitoba on March 20[th] 2020. Food and Alcohol services were restricted to facilities with 50 or fewer people or 50% capacity, whichever was lower, effective that same date [16]. Restaurants were declared shut down as of April 1[st] 2020 [17]. Schools were declared to be shut down on March 23[rd] 2020. Although differences exist in effective person-to-person contacts and the current stage of the pandemic around the world, our hope is that our study can help inform policy not only in Manitoba, but globally.

## Methods

From publicly available sources [18,19], we obtained the number of confirmed cases, number recovered, and deaths, from March 12 (the date of the first confirmed case in Manitoba) until May 4. In Manitoba, public health authorities permitted the first relaxation of social distancing behaviours beginning May 4.

### Model structure

We developed a deterministic compartmental model, in which we divided the population into five compartments based on infection and self-isolating behavior status. Details are provided (S1 Appendix). In brief, this is a modified Susceptible->Infected->Recovered (SIR) model, in which the infected people are partitioned into three: (1) those with a propensity to self isolate but are not yet self-isolating, (2) those who will not self-isolate throughout their infectious period, and (3) those who are currently self-isolating. Self isolation may be affected by a number of factors, including public health responses [20] and awareness of ones own infection status. Estimates of asymptomatic infections may be as high as 18% [21], or even 31% [22]. We think of those with "a propensity to self isolate but are not yet self-isolating" as people who will self-isolate once their infection status becomes known but who are not yet symptomatic (i.e., are infectious but are in their pre-symptomatic period) or for other reasons do not yet know their status. We think of those who will not self-isolate throughout their infectious period as people who either never become symptomatic, or who disregard their symptoms and continue

to contact others. Regardless of the reason for self-isolation or non-self-isolation, the model accounts for these groups of people by removing those who are currently self-isolating from the population of infected people who may infect others.

Using the Runge-Kutta 4 integration method with 0.2 of a day step sizes in the Berkeley Madonna software [23], we ran the fitted model for five years from the date that relaxation of social distancing regulations began in Manitoba, but endpoints–time until peak prevalence and time until near elimination (< one case per million)–were reached in almost all scenarios in less than three years. Because we examined relatively short-term impacts of different social distancing scenarios, we ignored births and non-COVID-19 deaths in the model. The original model structure for our work is not age stratified, which implies a homogenously mixing population. While this may be unrealistic in the long run, most of the potential effects of heterogenous mixing by age will be muted while young people are home schooled or on holidays. Also, because the model fit the early observed data (to May 4) quite well, we decided to keep the model as simple as possible, but no simpler; for example, the age-mixing matrices are unknown and there was little evidence whether infectiousness or susceptibility of young people differs from that of older people. In the time since our original work was conducted, there has been more evidence regarding possible differences in susceptibility between ages [24,25]. To assess the impact that non-homogenous mixing patterns and differences in susceptibility between ages may have on our results, we updated our model to include two age groups with different COVID-19 susceptibility and with heterogenous mixing patterns between ages. We compared the updated model results with our original results.

### Model parameterization

We fit the model to data beginning on day 15 (March 27) of the epidemic in Manitoba. This minimized the effect of testing scale-up on empirically estimated patterns of new cases. The nearly linear rise in cumulative tests within two weeks of the first test (Fig 1) suggests that within two weeks of the beginning of the epidemic in Manitoba, the magnitude of the number

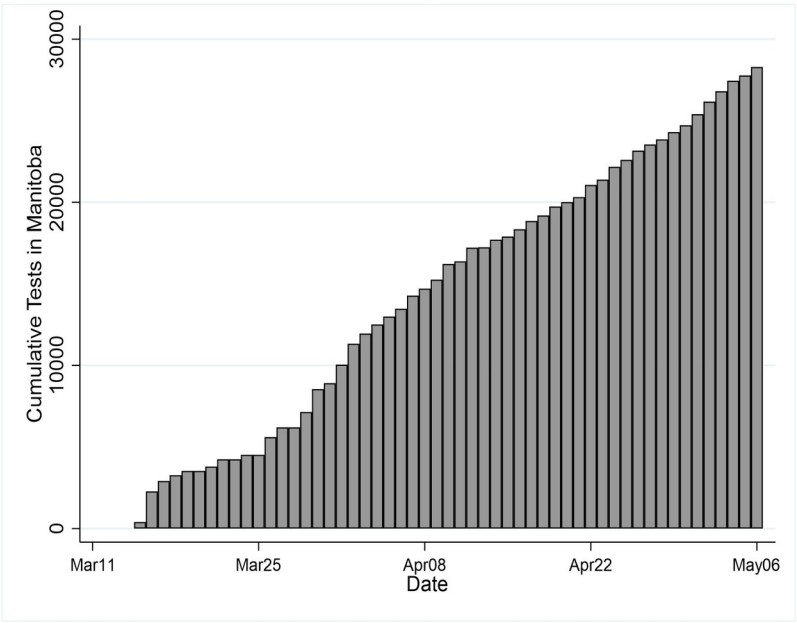

**Fig 1. Cumulative COVID-19 tests by day in Manitoba.**

of cases may be under-represented, but any pattern in the rise and fall of actual cases is likely to resemble closely the rise and fall in observed (test-confirmed) cases.

To fit the model to empirical data, we created 30,000 sets of parameter values using Latin Hypercube Sampling [26]. Each set consisted of values sampled uniformly from the plausible ranges of seven model fitting parameters (Table 1). Because evidence for COVID-19 parameter values is still scarce, plausible ranges for the parameters were kept wide. With each set of parameter values, we used the model to estimate the epidemic. The goodness of fit (GoF) for a given set of parameter values was determined by summing the squared difference between the empirical and model-estimated number of cases, people recovered, and deaths, for each day, beginning on Day 15 of the epidemic in Manitoba. The fits with the lowest GoFs were the best fitting sets of values.

To assess the impact on results of the under-reporting of cases, we conducted the above fitting process three times. We fit the model to: (i) unaltered empirical data, (ii) modified data assuming that 2/3 of cases are reported, so with the observed death counts but with cases and recovered counts increased by 50%, and (iii) modified data assuming 66% under reporting, so with observed death counts but with cases and recovered counts increased by 300%. We selected 36 sets of best-fitting parameter sets–the 12 sets with the best GoF for each of the three fitting processes—and examined epidemic trends for each set of best-fitting parameter values. In this way, we report a range of expected epidemic outcomes.

## Basic reproductive number (R0)

R0 for COVID-19 has been estimated with values ranging from 2.0 to 14.5 [10,27]. R0 is defined as the number of new infections that a single infectious person would infect, on average, in a totally susceptible population. COVID-19 is no different from other infections, in that R0 will vary from setting to setting. R0 is determined by infectiousness, infectious duration, and the degree to which people encounter each other in a manner sufficient for infection transmission. The degree of effective contacts between people may differ geographically due, for example, to culture or population density–as well as changes due to public health directions for social distancing.

The effective reproductive number, Re(t), is defined as the number of new infections that an infectious person would infect, on average, in a population in which not everybody is susceptible. Re is a function of R0 times the fraction of population that remains susceptible. The most common way that Re may drop below one and thus trigger a decline in numbers of new cases, is by reducing the number of susceptible people (for example, through vaccination, or through a susceptible person becoming infected and no longer susceptible). In the case of COVID-19, the immediate response to the pandemic globally had a dramatic effect on Re by reducing contacts sufficient to transmit infection; that is, by directly reducing R0(t).

In this study, we account for social distancing by allowing R0 in the model to decrease. We chose to model R0 directly, rather than have separate parameters for infectiousness and for contacts, because we could then use the literature on estimated values of R0 to inform the range of R0 parameter values that we used for model fitting. However, if we assume that infectiousness is constant for a given population, then a reduction in R0 by a given proportion would be the same as a reduction in contacts by the same proportion. After allowing R0 in the model to decrease, we then examine the impact of social distancing relaxation on the course of the epidemic by allowing R0 in the model to rise after the date at which social distancing relaxation policies were initiated. We thus examine 4 different policy changes: "hold the course" with no relaxation, small relaxation of social distancing (return R0 to 15% of pre-epidemic R0), medium relaxation (return R0 to 25% of pre-epidemic R0), and large relaxation (return to 50% of pre-epidemic R0).

**Table 1. Parameter value ranges used in model fitting and the values in the best fitting models.**

| | R0(0)[1] | Cut Rate[2] | Dur Infect[3] | SI start[4] | Prop SI[5] | Time to SI[6] | Case Fatality[7] |
|---|---|---|---|---|---|---|---|
| Range[8] | 1.5–15.0 | 0.900–1.000 | 8.0–28.0 | 10.0–60.0 | 0.30–1.00 | 1.0–10.0 | 0.010–0.070 |
| Best fitting parameter value sets when fitting to unaltered empirical data | | | | | | | |
| 1 | 8.1 | 0.930 | 14.4 | 13.3 | 0.36 | 4.9 | 0.040 |
| 2 | 7.8 | 0.958 | 18.4 | 12.1 | 0.89 | 6.7 | 0.028 |
| 3 | 9.3 | 0.927 | 17.4 | 28.7 | 0.80 | 7.2 | 0.028 |
| 4 | 10.8 | 0.934 | 20.5 | 16.3 | 0.41 | 9.5 | 0.029 |
| 5 | 9.9 | 0.930 | 18.7 | 23.2 | 0.37 | 5.6 | 0.040 |
| 6 | 14.2 | 0.908 | 22.7 | 30.7 | 0.76 | 2.7 | 0.038 |
| 7 | 12.9 | 0.912 | 20.3 | 42.2 | 0.61 | 9.2 | 0.031 |
| 8 | 12.4 | 0.922 | 21.1 | 25.1 | 0.68 | 9.0 | 0.028 |
| 9 | 10.5 | 0.926 | 19.9 | 58.0 | 0.47 | 3.9 | 0.032 |
| 10 | 9.8 | 0.931 | 18.4 | 21.6 | 0.35 | 6.9 | 0.025 |
| 11 | 9.9 | 0.926 | 19.1 | 57.0 | 0.45 | 4.6 | 0.034 |
| 12 | 8.3 | 0.945 | 17.5 | 28.7 | 0.34 | 8.7 | 0.028 |
| Mean (Range) | 10.3 (7.8–14.3) | 0.929 (0.908–0.958) | 19.0 (14.4–22.7) | 29.7 (12.1–58.0) | 0.54 (0.34–0.89) | 6.6 (2.7–9.5) | 0.032 (0.025–0.040) |
| Best fitting parameter value sets when fitting to unaltered deaths, but 3/2 times number of cases and recovereds | | | | | | | |
| 1 | 7.5 | 0.933 | 12.6 | 49.0 | 0.79 | 2.4 | 0.031 |
| 2 | 8.9 | 0.940 | 17.4 | 21.6 | 0.55 | 9.1 | 0.023 |
| 3 | 8.3 | 0.956 | 18.3 | 15.3 | 0.86 | 3.6 | 0.024 |
| 4 | 8.4 | 0.936 | 15.4 | 15.3 | 0.35 | 2.2 | 0.018 |
| 5 | 13.8 | 0.919 | 22.6 | 22.1 | 0.40 | 9.5 | 0.029 |
| 6 | 12.5 | 0.928 | 22.5 | 24.5 | 0.99 | 1.2 | 0.015 |
| 7 | 14.4 | 0.922 | 21.7 | 10.4 | 0.56 | 2.8 | 0.018 |
| 8 | 8.7 | 0.938 | 16.9 | 24.8 | 0.56 | 7.9 | 0.015 |
| 9 | 9.8 | 0.928 | 17.1 | 33.3 | 0.71 | 7.3 | 0.015 |
| 10 | 9.6 | 0.928 | 16.9 | 19.3 | 0.57 | 7.0 | 0.021 |
| 11 | 9.7 | 0.940 | 19.3 | 20.7 | 0.52 | 8.3 | 0.016 |
| 12 | 6.6 | 0.945 | 14.1 | 37.8 | 0.95 | 2.8 | 0.020 |
| Mean | 10.1 (7.5–14.4) | 0.933 (0.919–0.956) | 18.3 (12.6–22.6) | 23.3 (10.4–49.0) | 0.62 (0.35–0.99) | 5.6 (1.2–9.5) | 0.020 (0.015–0.031) |
| Best fitting parameter value sets when fitting to unaltered deaths, but 3 times number of cases and recovereds | | | | | | | |
| 1 | 8.6 | 0.950 | 17.6 | 35.7 | 0.81 | 7.2 | 0.010 |
| 2 | 9.4 | 0.944 | 16.6 | 54.1 | 0.66 | 4.1 | 0.014 |
| 3 | 8.3 | 0.946 | 16.8 | 41.5 | 0.54 | 6.3 | 0.014 |
| 4 | 6.5 | 0.952 | 12.1 | 48.0 | 0.76 | 4.3 | 0.010 |
| 5 | 7.9 | 0.963 | 17.3 | 21.2 | 0.72 | 6.0 | 0.013 |
| 6 | 12.1 | 0.919 | 17.5 | 32.3 | 0.36 | 5.2 | 0.012 |
| 7 | 10.2 | 0.933 | 15.7 | 19.1 | 0.51 | 3.0 | 0.011 |
| 8 | 14.0 | 0.911 | 18.0 | 19.3 | 0.83 | 7.2 | 0.012 |
| 9 | 8.8 | 0.938 | 15.0 | 28.0 | 0.98 | 5.0 | 0.012 |
| 10 | 7.8 | 0.952 | 15.3 | 22.8 | 0.94 | 8.7 | 0.011 |
| 11 | 8.8 | 0.939 | 15.0 | 38.6 | 0.95 | 4.8 | 0.011 |
| 12 | 9.7 | 0.946 | 18.9 | 17.7 | 0.70 | 9.0 | 0.012 |
| Mean | 9.4 (6.5–14.0) | 0.941 (0.911–0.963) | 16.3 (12.1–18.9) | 31.5 (17.7–54.1) | 0.73 (0.36–0.98) | 5.9 (3.0–9.0) | 0.012 (0.010–0.014) |

[1] R 0(0) at the beginning of the epidemic. In our modelling work, R0(t) changed with time to reflect changes in social distancing. Re(t) was then determined as R0(t) * (the proportion of the population still susceptible).

[2] Daily proportion by which R0(t) was cut due to social distancing (prior to social distancing relaxation).

[3] Average duration of SARS-CoV-2 infectiousness.

[4] Time (days) from epidemic start until some infected people began to self isolate.

[5] Proportion of infected people with a propensity to self isolate.

[6] Average time from infection until self isolate among infected people with a propensity to self isolate.

[7] Case fatality proportion.

[8] Range used in Latin Hypercube Sampling to obtain the 30,000 parameter value sets for model fitting.

## Descriptive analysis

For each of the 36 best-fitting sets of parameter values, we projected model time forward by five years and recorded five outcomes: (1) time until near elimination of COVID-19 (which we defined as < one case per million), (b) time until peak prevalence, (c) total proportion of the population affected within one year, (d) value of peak prevalence, and (e) total deaths within one year. We obtained these five measures under four conditions of social distance relaxation and three ranges for the proportion of infected people with a propensity to self isolate (who will self isolate sometime after infection, at a specified rate of self isolation). The four scenarios of social distance relaxation were: (1) no relaxation, i.e., R0 remaining at the lowest level that it reached before relaxation of social distancing regulations was initiated, (2) relaxation to 15% of what it was pre-COVID-19, (3) relaxation to 25% of what it was pre-COVID-19, and (4) relaxation to 50% of what it was pre-COVID-19. Although it is not too late to reverse it, social distancing relaxation in Manitoba, as in much of the world, has already begun. We neverthe-less included projections with no relaxation as a base scenario to which the other scenarios could be compared. The three scenario ranges for the percent of infected people with a pro-pensity to self-isolate were: (1) 30–50% (some self-isolating), (2) 50–80% (most self-isolating), and (3) > 80% (nearly all self-isolating).

We used medians and interquartile ranges (IQR) to describe the range of modelling results for each of the five epidemic outcomes, under each of the four social distancing relaxation sce-narios and three self-isolating ranges. Where we used box and whisker plots to present our findings, the box corresponds to the IQR and whiskers and dots display the full range of the data, following Tukey guidelines [28].

# Results

## Model fitting

As seen in Fig 2, the 36 best-fitting sets of parameter values result in models that fit very well to the unaltered observed values of both deaths and recovered individuals. Though the models do not quite capture the steep rise in cases around day 25 of the epidemic, they do capture the more general pattern of rise and fall in the number of cases from day 0 (March 12) to day 56 (May 4). Similarly, as a sensitivity analysis on the effect of incomplete reporting of cases, visual inspection of the 12 best fits to our two altered data sets suggest very good fits to deaths and recovered individuals and good fits to numbers of cases. The pattern (rise and fall) of the num-ber of cases and recovered individuals, was the same for all three data sets (observed and altered); only the magnitude of cases and recovered individuals differed. Recall that the num-ber of deaths was not altered in the two modified datasets.

The empirically estimated number of cases who recovered on April 8 in Manitoba (day 28 in the model) was 48. The cumulative number of recovered people rose from 21 on April 7 to 69 on April 8 [18,19]. This empirically estimated rise in recovered people (and corresponding drop in active cases) may partly be an effect of delayed reporting of recovered cases and may partly explain the inability of any of the model fits to entirely replicate the sharp rise in cases in the few days prior to day 28, and the sharp drop on day 28.

All 36 best fitting sets of parameter values resulted in estimates of R0 less than one on the day before the relaxation of social distancing regulations began in Manitoba. The range of R0 in the best fitting sets was 6.5–14.4 on day one of the epidemic, falling to 1.4–3.1 by day 25 and to 0.076–0.965 by day 56 (the day that social distancing relaxation began). This represents a reduction in R0 of 88–99% from day one to day 56 of the epidemic in Manitoba. That is, R0 was at a level of 1–12% of its baseline level on the day before relaxation in social distancing was

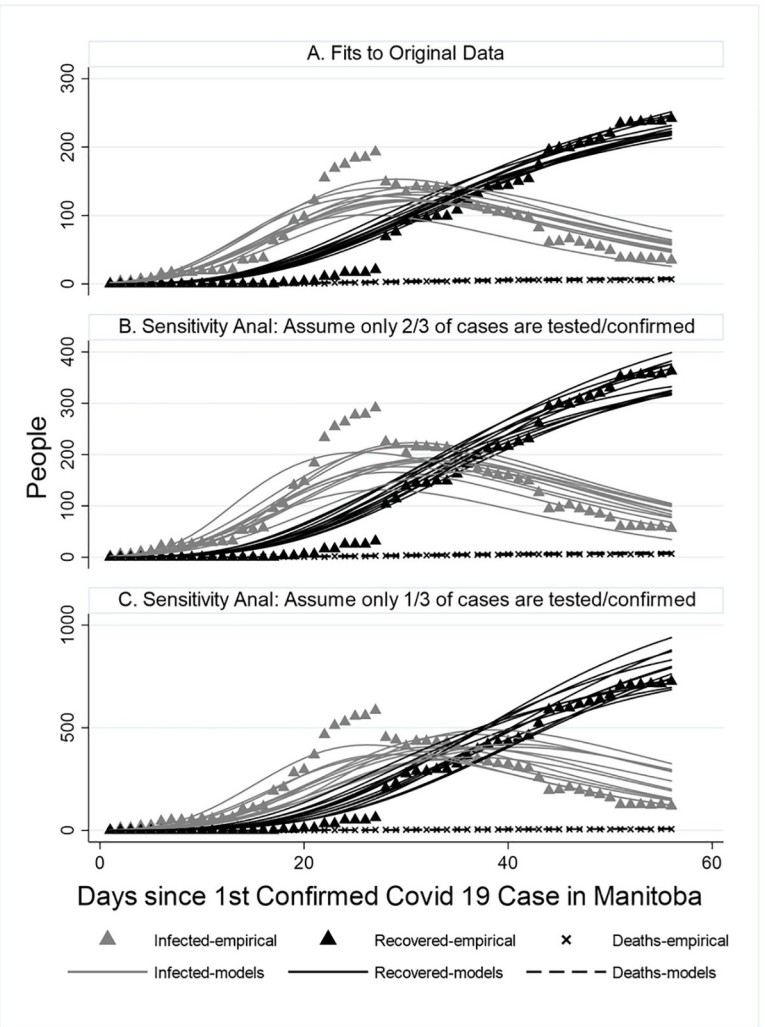

**Fig 2. Model fitting.**

permitted. Our "hold the course" projection thus reflects this level, and our small (15%), medium (25%) and large relaxation (50%) of social distancing should be viewed in this context.

## Descriptive modelling results

Other than expected differences in the estimated mortality rates, the version of empirical data used in fitting (unaltered or altered) did not affect how social distancing relaxation and self-isolation after infection influenced the epidemiologic trends. Hence, to simplify interpretation, we have pooled the 3 sets of analyses (i.e., analyses of the original data and of the two altered datasets) and summarize results from all 36 sets of parameter values in the next sections.

## Impact of social distance relaxation on the expected extent of the COVID-19 epidemic

With no social distancing relaxation, our study suggests that COVID-19 near-elimination could occur within four to six months of the beginning of the epidemic in Manitoba (March

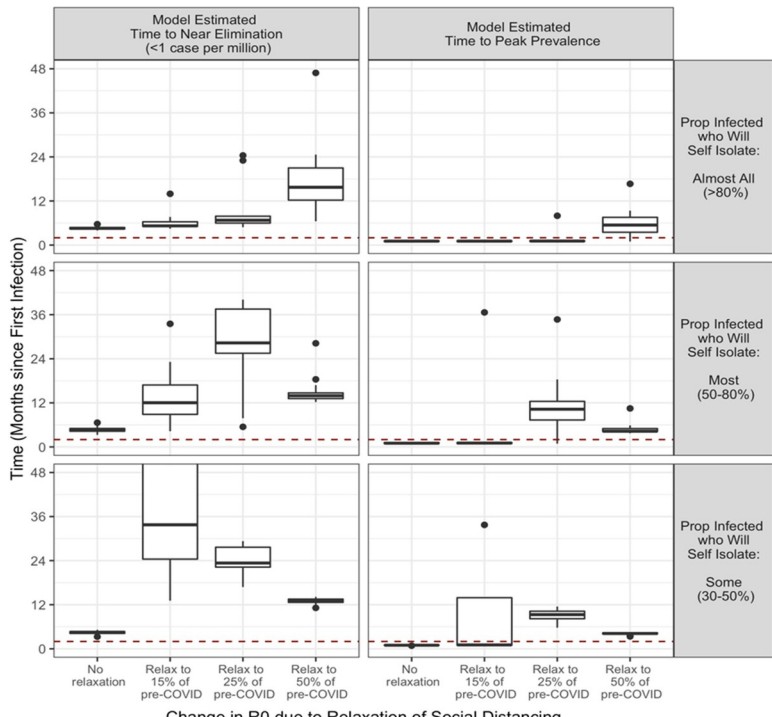

**Fig 3. Model estimated months to COVID-19 near elimination and peak prevalence, stratified by proportion of infected who will self-isolate and degree of social distance relaxation.**

12), in all modelled scenarios about the percent of infected people who will self isolate (Fig 3). Relaxing social distancing to contact levels that are just 15% of what they were prior to COVID-19 resulted in a median time until near-elimination of 7 months, 12 months, and 2.9 years, depending on whether > 80%, 50–80%, or 30–50% of infected people self-isolate, respectively (Fig 3). The time until near elimination is estimated to climb further if social distance relaxation results in contact levels that are 25% of what they were prior to COVID-19, but will drop in most cases if contact levels rise to 50% of what they were prior to COVID-19 (median time to near elimination 16 months if > 80% of infected people self isolate, but 13.5 months and 13 months if 50–80% of infected people self-isolate or 30–50% of people self-isolate, respectively).

The drop in time until near elimination of COVID-19 if contact levels rise to 50% of what they were prior to COVID-19 comes at a cost. In this scenario, many people will become infected within one year, with a median estimate of 32%, 88%, and 94%, depending on the proportion of the population who self isolate (Fig 4, right panel). In addition, they will become infected fast (Fig 3, right panel). Indeed, at this amount of social distance relaxation, peak prevalence will likely occur within months, possibly before this paper is published, and the peak prevalence will be very high, with estimates of 31–38% of the population in many scenarios (Fig 4, Model Estimated Peak Prevalence panel). Assuming that those who have recovered are effectively immune to re-infection, this means the number of new infections caused by one infected individual, Re(t), will decrease as the proportion of the population remaining susceptible rapidly approaches zero. In other words, in these scenarios, the epidemic will flare quickly and then burn out.

Young people may be less susceptible to COVID-19 than older people, and mixing patterns may differ by age. Using our same model but updated with age stratification, we found that the population-level results described above are generally unaltered. Peak prevalence and time

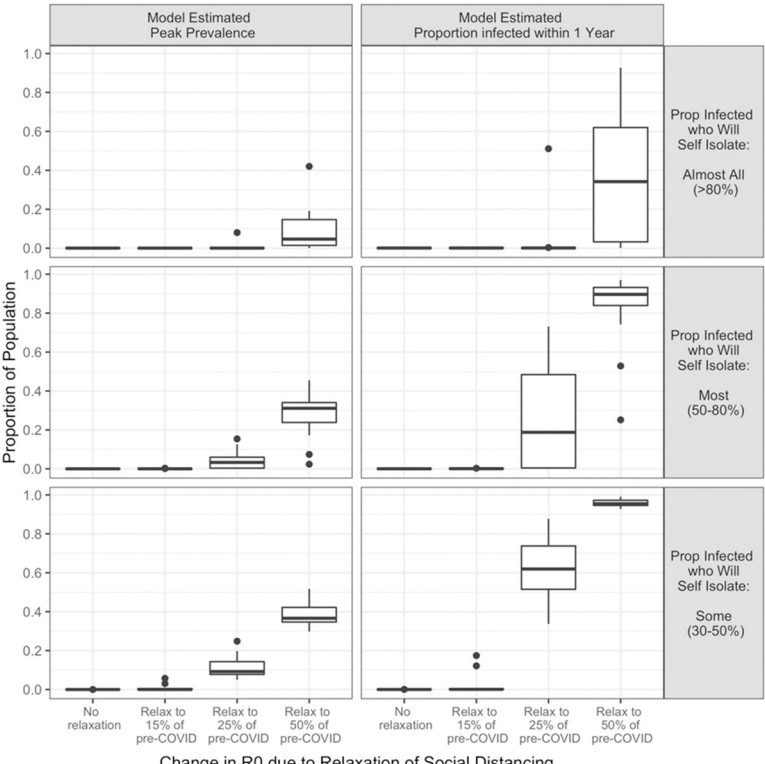

**Fig 4. Model estimated peak prevalence and proportion infected within one year, stratified by proportion of infected who will self-isolate and degree of social distance relaxation.**

until peak prevalence may vary substantially by age, however. See S2 Appendix for details of these results.

## Impact of social distance relaxation on deaths

Relaxation could increase the expected number of deaths due to COVID-19 in the coming year in Manitoba from very few ($< 15$) to many ($> 30{,}000$, Table 2). In scenarios with a

**Table 2. Model estimated number of deaths due to COVID-19 in Manitoba during the first year of the epidemic (population ~1.36 million): Median (IQR).**

| Behavior adjustment resulting from social distance relaxation | Proportion of those infected who will self isolate | | |
|---|---|---|---|
| | 0.3–0.5 | 0.5–0.8 | > 0.8 |
| No relaxation ("hold the course")[1] | 9 (9–10) | 10 (8–12) | 9 (8–11) |
| Small Relaxation: R0 increases to 15% of what it was pre COVID-19[2] | 202 (18–1,332) | 19 (12–42) | 11 (10–15) |
| Medium Relaxation: R0 increases to 25% of pre COVID-19[2] | 22,893 (7,735–34,431) | 5,856 (379–17,546) | 17 (11–42) |
| Large Relaxation: R0 increases to 50% of what it was pre covid-19[2] | 37,451 (24,608–48,685) | 21,502 (9,307–38,274) | 5,667 (706–12,549) |

[1] On Day 56 of the epidemic in Manitoba, when social distance relaxation began, R0 was estimated to be $< 1$ in all best-fitting sets of modelled parameters values (range of R0 values by day 56: 0.07–0.97, for a reduction in R0 since day 1 of the epidemic of 88–99%).

[2] Pre-COVID (or day 1 of the epidemic), model estimated R0 in the 36 best fitting sets of modelled parameter values ranged from 6.5 to 14.4.

reasonable assumption about the extent of self-isolation achievable among infected people (50–80%), relaxing social distancing may result in a median of 19 deaths (IQR 12–42), 5,856 deaths (IQR 379–17,546), or 21,502 deaths (IQR 9,307–38,274), depending on the degree of social distancing relaxation. The wide IQR for each estimate is due primarily to differences in estimated case fatality in the best fitting scenarios. Case fatality estimates ranged from 1%-4%. In the scenario that estimated 37,451 deaths within a year (median estimated deaths when 30–50% of people self isolated and social distancing relaxed to 50% of pre-COVID-19 levels), estimated case fatality was 2.9% and estimated proportion of the population infected within a year was 94.9%.

## Discussion

The immediate global response to the COVID-19 pandemic by social distancing likely resulted in a decline in R0 to < one in much of the world. This early response has given health care workers some needed time to prepare. The degree of social distance relaxation, however, will impact both how much longer we have until peak COVID-19 prevalence, and how high the peak prevalence will be. Flattening the epidemic curve is crucial for two reasons: (1) it lengthens the time until peak prevalence allowing more preparation time, and (2) it lowers the ultimate peak prevalence allowing us to meet the needs of infected people with fewer resources as the resource utilization is not required all at once.

Our aim was not to provide exact estimates of people who would become infected or people who would die in Manitoba, or in regions of the world similar to Manitoba. There are too many variables to estimate this with any accuracy. For example, what would be the impact of reducing full contacts but maintaining many partial contacts in which infection is possible but less likely (e.g., reducing prolonged contacts between people)? Among people who self-isolate, what proportion still have some contacts, even if only partial (e.g., sharing a kitchen or bathroom with household members)? Our goal was therefore to explore the plausible impact of different degrees of social distancing relaxation, while holding constant other factors that may impact the spread of COVID-19. That is, the relative differences in our modelled outcomes between scenarios of different social distancing relaxation should be the focus of interpreting our results, rather than the exact outcome.

Our results suggest that even if we relax social distancing so that levels of contact return to just 25% of what they were prior to COVID-19, and if we are able to achieve 50–80% self isolation of infected people within days of infection, we may expect to experience peak prevalence of infection within approximately 10 months of first case (which was March 12 in Manitoba) (IQR 8–12 months), and that the peak prevalence may be 4% of the population (IQR 1–8%). To put this in context, a peak prevalence of 4% of the population of Manitoba would mean 54,400 people infected at the same time. Not all will require medical care, but current literature suggests that 10% of COVID-19 cases may require medical support (e.g., hospitalization or ICU) [29]. If 10% require medical support, that is > 5,000 people. Relaxing social distancing to levels of contact that are 50% of what they were prior to COVID-19 may, according to our estimates, result in over 35% of the population infected at the same time.

Our results suggested that if we held the course, that is, not let up on social distancing, we may have achieved near elimination of this coronavirus within 4–6 months of the first confirmed case (March 12 in Manitoba). Indeed, since this study was conducted in May 2020, the number of confirmed active cases of COVID-19 in Manitoba fell to 3 people by May 22 and 1 person by July 13, but as social distancing relaxation continues, the number of active cases has started to rise and was at 74 cases by July 27 [30]. A caveat to this estimate is that each local epidemic is not a closed population. COVID-19 is in a pandemic state. Thus, maintaining strict

social distancing in one geographic region may not eradicate COVID-19 from that region if travellers from other regions bring in new infections. That said, if policy in a geographic region is to hold the course in terms of continued social distancing, then new infections that may be brought in from outside of the region will not spark local infections.

If holding the course is unachievable, our results suggest that a very small relaxation of social distancing (to a level of contacts that is 15% of what it was pre-COVID-19) may extend the epidemic to years. However, although the duration of the epidemic will be significantly extended, the curve will be flattened compared to no social distancing, reducing the number of people requiring medical care at the same time, and thus allowing health care workers to provide better care for those requiring it.

The bulk of our modelling work assumed a homogenously mixing population. This assumption was sufficient for fitting the model quite well to empirical data. While schools were closed, in fact, this may have been a reasonable assumption. With the opening of schools, however, age stratified mixing may become an important factor in local epidemics. As has been seen in other infectious diseases, intense mixing between children during school periods, followed by less mixing during holidays, may lead to cyclical patterns of new COVID-19 cases [6,31,32]. In our updated model, stratified by age, our population level results were very similar to those from our original work. However, young people (< age 20) may reach a much lower peak prevalence than older people, and may have a different timeline until peak prevalence. For this reason, age-stratified modelling will help to inform policy about the plausible effectiveness of interventions aimed at specific age groups.

It may seem counter-intuitive that the parameter value sets that fit well to unaltered empirical data gave similar conclusions about the impact of social distancing relaxation on the extent of the epidemic as the parameter value sets that fit well to data that was augmented by a factor of 3/2 and three, respectively. However, although the number of cases and recovered people in the altered data were higher than the number in the original data, the pattern in the number of cases and recovered people (that is, the rise and fall over time) was the same. We believe that this is a reasonable assumption, given that the trend in numbers of new tests was linear after approximately two weeks of the epidemic in Manitoba. In all three fittings, the number of new cases was falling just prior to the initiation of social distancing relaxation. This means that Re was < one. Re is a function of both R0 and the proportion of the population that is susceptible. At this early stage of the epidemic, nearly everybody is still susceptible, so the main factor influencing Re is the changing R0 (due to increased social distancing, such as temporary closing of businesses and schools, in response to the epidemic). As Re was < one just prior to social distance relaxation in all three versions of the data, it is reasonable that the impact of increasing R0 (and therefore increasing Re) would be similar across all three versions of the fitted data.

Our estimate of the case fatality varied from approximately 1% to 4% in the different model fits. The primary factor influencing the case fatality estimate was which of the three versions of empirical data we fit to. In two of the versions, we increased the number of cases and recovered people by a factor of 3/2 and three times the original empirical data, in order to explore the impact that under-recording of COVID-19 cases may have on our results. However, the number of deaths due to COVID-19 was constant in all three versions of empirical data. Clearly, this led to model estimated case fatality among the altered empirical data sets that were lower than that found when fitting to the original empirical data set. Since the time of our original modelling work, there has been evidence of potential under-reporting of COVID-19 deaths [33–35]. It is possible, therefore, that the actual number of COVID-19 deaths may be under-represented in the empirical data that we fit to. If so, this would lead to an underestimate of mortality in our modelling. Conversely, our best fitting model estimated case fatality rates

were within range but somewhat higher than that estimated by Abdollahi et al [36], who estimated an adjusted case fatality rate in Canada (adjusting for under-reporting of cases, and lag time from infection to death) of 1.6% (credible interval 0.7%-3.1%). If case fatality is 1.6%, then the median estimated deaths in our scenarios of relaxation of social distancing to 50% of pre-COVID-19 contacts would be approximately: 20,650 (30–50% self-isolation among those infected), 18,931 (50–80% self-isolation), and 6,745 (>80% self-isolation).

It is possible that younger people have a different susceptibility to COVID-19 than older people [37]. There have also been suggestions that COVID-19 infection may not confer lifetime immunity to reinfection [38]. Neither of these aspects of the epidemic were incorporated in the original modelling work, but as more knowledge has since become available, we did compare our original results with results from an updated age-stratified model.

Although policy in the form of mandatory social distancing strongly influenced behaviour at the beginning of the epidemic, voluntary reductions in contacts resulting from perceived risk likely plays a role at the current stage [39,40]. Perceived risk of COVID-19 is likely influenced by the number of new infections. When the number of new infections was low, such as in Manitoba when we had reached only one active case in the middle of July, policy would have a larger impact on social distancing. As perceived risk increases, as it has with the rising numbers of cases in Manitoba, voluntary reductions in contacts may play as much of a role in containing the epidemic as policy mandated social distancing.

## Limitations

Our results apply to a population without access to a vaccine. Numerous COVID-19 vaccine candidates are currently being explored [41,42]. If we manage to maintain a low level of infections in the population for long enough, vaccine availability will ultimately alter the conclusions of this study about the expected time until near elimination of COVID-19. The results in this paper are at the population-level; that is, pooled across all ages. Our modelled outcomes (e.g., estimated peak prevalence) will likely vary by age. We also note that self-isolation behaviours and social distancing regulations may well be correlated, and we have not attempted to model any such relationship. This correlation could lead to under- or over-estimation of the number of infections, depending on the direction and strength of this association. Further work would be of value when such behavioural data become available.

## Conclusions

Policy makers and healthcare planners need to be aware that even a small relaxation of social distancing (even to a level of contacts that is still much lower than it was pre-COVID-19) may immensely impact both the pandemic duration, and the proportion of the population affected. Relaxation to 50% of pre-COVID-19 contacts may result in >90% of the population affected, and a peak prevalence within months of > 35%. Only holding the course with respect to social distancing may have resulted in near elimination of COVID-19 by Fall 2020, while relaxing social distancing to 15% of pre-COVID-19 contacts will flatten the curve but may greatly extend (to years) the duration of the pandemic.

## Supporting information

**S1 Appendix. Model description.**
(DOCX)

**S2 Appendix. Modelling two age groups.**
(PDF)

## Acknowledgments

We thank the members of the Centre for Healthcare Innovation's Biostatistics Brainstorming Group in Winnipeg for helping to shape this study.

## Author Contributions

**Conceptualization:** Leigh Anne Shafer, Marcello Nesca, Robert Balshaw.

**Data curation:** Leigh Anne Shafer.

**Formal analysis:** Leigh Anne Shafer.

**Methodology:** Leigh Anne Shafer, Robert Balshaw.

**Software:** Marcello Nesca.

**Supervision:** Leigh Anne Shafer.

**Writing – original draft:** Leigh Anne Shafer.

**Writing – review & editing:** Leigh Anne Shafer, Marcello Nesca, Robert Balshaw.

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
