## [Decision Letter · Decision Letter 0]

1 Oct 2020

PONE-D-20-23700

Hold the Course and Flatten the Curve: Model Estimated Impact of Decisions to Relax Social Distancing on COVID-19 Epidemic in Manitoba, Canada

PLOS ONE

Dear Dr. Shafer,

Thank you for submitting your manuscript to PLOS ONE. After careful consideration, we feel that it has merit but does not fully meet PLOS ONE’s publication criteria as it currently stands. Therefore, we invite you to submit a revised version of the manuscript that addresses the points raised during the review process.

Please respond to the reviewer comments on a point-by-point basis and revise the manuscript accordingly. 

We look forward to receiving your revised manuscript.

Kind regards,

Jeffrey Shaman

Academic Editor

PLOS ONE

Journal Requirements:

Reviewers' comments:

Reviewer's Responses to Questions

**Comments to the Author**

1. Is the manuscript technically sound, and do the data support the conclusions?

Reviewer #1: Yes

Reviewer #2: Yes

2. Has the statistical analysis been performed appropriately and rigorously? 

Reviewer #1: N/A

Reviewer #2: Yes

3. Have the authors made all data underlying the findings in their manuscript fully available?

Reviewer #1: Yes

Reviewer #2: Yes

4. Is the manuscript presented in an intelligible fashion and written in standard English?

Reviewer #1: Yes

Reviewer #2: Yes

5. Review Comments to the Author

Reviewer #1: Referee Report: “Hold the Course and Flatten the Curve: Model Estimated Impact of Decisions to Relax Social Distancing on COVID-19 Epidemic in Manitoba, Canada” (PONE-D-20-23700)

Summary

In this paper the authors use a compartmented model of the COVID-19 pandemic to understand how the epidemic would evolve in a single area. The paper is, for the most part, clear and appears to be well-executed (although some of these methods are outside my usual area of expertise)

Positives

Use of a single, contained geographic region provides for a clear exposition of how the epidemic is evolving over time and may reduce the need to worry about mobility as a potential new source of infections in the model.

Major items

I have several concerns, many of which likely arose in the gap between when the authors wrote the paper and the time at which I am reading it.

Two of the biggest issues to my mind are:

1. Homogenous mixing. This assumption may be plausible during the lockdown period, but once the lockdown has eased it no longer seems like a good assumption, especially since there is some evidence that the transmissibility of COVID-19 varies by age. I would (ideally) like to see the authors incorporate some age structure into their model, but I think a careful discussion of the biases that homogeneous mixing brings in would be sufficient.

2. Perfect self-isolation. The authors assume that, conditional on choosing to self-isolate, self-isolation is perfect. Given the ways in which the authors break out the three types of infected individuals I can appreciate that this issue may seem odd, but even among people who are “perfectly” self-isolating, there may be some risk of contact with others. Again I think that this is most likely a discussion point (much like the behavioral response I discuss below).

Interdisciplinary issues. Reading this as an economist I am struck by the way in which R0 is treated. On the one hand, it seems to be what I would call a “structural parameter”—something that is invariant to the world around it—but it appears in this case to not have a structural interpretation but rather be the product of a structural parameter and a measure of the frequency of contacts. It would be helpful if the authors could address this issue because it would make the adjustment that they make to R0 seem less ad hoc. In particular, the reduction in R0 during the lockdown should be related to observed changes in physical interactions that could transmit COVID-19. I would like to see the changes in R0 compared to some measure of how mobility changed in Manitoba during (and before) the lockdown.

There is also a behavioral response that is missing in this paper, although my understanding of this literature is that behavioral responses are typically ignored. These responses may arise because people endogenously choose to reduce contacts in response to a perceived greater risk of infection. In the cases of the models that the authors present, that would mean that R0 itself would depend on the number of new infections and is (presumably) a decreasing function of the number of new infections. Examples of this phenomenon can be observed in any of the myriad studies (e.g. Andersen, 2020; Gupta et al., 2020) using cellular data to study changes in mobility in the United States (I appreciate that isn’t Canada, but the idea of voluntary distancing is likely to be universal).

Minor items

Descriptions in text and figures are sometimes in reverse order. Please use a consistent order so that the reader knows how the discussion and figures relate more easily (i.e. figure III and line 153).

It would be helpful it the authors explicitly noted that their model used day for the time period.

What is happening to the empirical infections around day 28? There is a very large decrease. I would like to have some discussion of this—did Manitoba change some policy measure?

It would be useful to rescale deaths so that they are visible on the same axis as the other measures in figure 2.

The authors could use their results to discuss the “herd immunity” controversy since that appears to be implicitly what is happening in their least distanced model when people do not self-isolate. One point that would be worth emphasizing is that the faster elimination of the epidemic comes at the cost of significantly more deaths. The authors do discuss this point, but I would encourage them to be even more explicit and put this into a cost-benefit framing.

References

Andersen, M. (2020). Early Evidence on Social Distancing in Response to COVID-19 in the United States (SSRN Scholarly Paper ID 3569368). Social Science Research Network. https://doi.org/10.2139/ssrn.3569368

Gupta, S., Nguyen, T. D., Rojas, F. L., Raman, S., Lee, B., Bento, A., Simon, K. I., & Wing, C. (2020). Tracking Public and Private Response to the COVID-19 Epidemic: Evidence from State and Local Government Actions (Working Paper No. 27027; Working Paper Series). National Bureau of Economic Research. https://doi.org/10.3386/w27027

Reviewer #2: Summary: This paper estimates the effects of relaxing social distancing restrictions on COVID-19 in Manitoba, Canada. The authors consider five outcomes (time until near elimination, time until peak, number of people infected within a year, share of population at peak, and total deaths within one year) and estimate how they would change under different social distancing relaxation levels. Using publicly available data and an SIR model, the paper finds that even a small degree of relaxing social distancing, up to 15% of the pre-COVID19 condition, could prolong the pandemic. A larger degree of relaxation (50%) could result in a fast and large peak in COVID-19 cases.

Contribution: This paper examines an important and timely issue. One contribution is that the authors look at both the timing and the size of the infection peak when social distancing is relaxing. The results could also inform policymakers about how a variation in relaxing social distancing policies could lead to vastly different outcomes of the pandemic.

Major Comments:

1. Given that the authors’ goal is to provide insights into the impact of relaxing social distancing in other parts of the world, it is essential to describe some of Manitoba’s characteristics such as population density, age, economic indicators, and average temperatures. These characteristics have been shown to correlate with the spread of COVID-19. Therefore, the results of this paper would be more relevant to places with similar characteristics.

2. The authors note that March 12 was the date of the first confirmed case and May 4 was the date of the first social distance relaxation. However, they did not mention the date that social distancing measures went into effect in Manitoba and how strict these measures were. It was not clear if this was considered in the model; however, the implementation of regulations can affect the spread rate, as found in many papers the authors cited. Furthermore, the timing between implementing and relaxing regulation could also affect how the data fit the model.

3. The definitions of social distancing relaxation levels were not clear. The authors define “no relaxation” as R0 “before the relaxation of social distancing regulations was initiated” (line 110), but the other levels (15%, 25%, and 50%) were defined as the percent of “what it was pre-COVID-19.” Although in footnote 2 under Table II, the authors note that “pre-COVID-19” represents day 1 of the epidemic in Manitoba, the table was mentioned in a much later part of the manuscript. I suggest having a dedicated description at the beginning.

4. For each outcome, the authors assume three different rates of self-isolation among infected individuals (30-50%, 50-80%, or >80%). Are the authors able to discuss how these levels would compare to the actual self-isolation rate? Also, self-isolation rates could correlate with social distancing regulations, which may result in over- or under-estimating. It is necessary to address this issue in the manuscript.

Minor Comments:

1. Line 10: it would be helpful to include the exact date of the statistics listed in the sentence “Since then, infections …”

2. Line 143 – 146: This paragraph may need additional explanation.

3. Line 167 – 171, starting with the sentence “The epidemic will therefore be nearly eliminated relatively quickly …” The use of the word “eliminate” can be misleading here. The authors refer to the period where the spread of the disease is low (i.e. R0<1) when 50% of social distancing is implemented. According to the authors, this happens when most of the population is infected. This scenario is only equivalent to elimination under the assumption that the possibility of re-infecting is zero. Although the authors, in the next paragraph, add that a 50% social distancing level also leads to faster and larger estimates of the peak, I suggest stating the assumption here or considering a different term.

6. PLOS authors have the option to publish the peer review history of their article (what does this mean?). If published, this will include your full peer review and any attached files.

Reviewer #1: **Yes: **Martin S Andersen

Reviewer #2: No

---

## [Author Response · Author response to Decision Letter 0]

14 Oct 2020

We have attached a separate document with a point by point response to reviewer comments.

---

## [Decision Letter · Decision Letter 1]

14 Dec 2020

Relaxation of Social Distancing Restrictions: Model Estimated Impact on COVID-19 Epidemic in Manitoba, Canada

PONE-D-20-23700R1

Dear Dr. Shafer,

We’re pleased to inform you that your manuscript has been judged scientifically suitable for publication and will be formally accepted for publication once it meets all outstanding technical requirements.

Kind regards,

Jeffrey Shaman

Academic Editor

PLOS ONE

Additional Editor Comments (optional):

Reviewers' comments:

Reviewer's Responses to Questions

**Comments to the Author**

1. If the authors have adequately addressed your comments raised in a previous round of review and you feel that this manuscript is now acceptable for publication, you may indicate that here to bypass the “Comments to the Author” section, enter your conflict of interest statement in the “Confidential to Editor” section, and submit your "Accept" recommendation.

Reviewer #1: All comments have been addressed

Reviewer #2: All comments have been addressed

2. Is the manuscript technically sound, and do the data support the conclusions?

Reviewer #1: (No Response)

Reviewer #2: Yes

3. Has the statistical analysis been performed appropriately and rigorously? 

Reviewer #1: N/A

Reviewer #2: Yes

4. Have the authors made all data underlying the findings in their manuscript fully available?

Reviewer #1: Yes

Reviewer #2: Yes

5. Is the manuscript presented in an intelligible fashion and written in standard English?

Reviewer #1: Yes

Reviewer #2: Yes

6. Review Comments to the Author

Reviewer #1: The authors have addressed my comments to my satisfaction. Thank you for the opportunity to review this paper.

Reviewer #2: (No Response)

7. PLOS authors have the option to publish the peer review history of their article (what does this mean?). If published, this will include your full peer review and any attached files.

Reviewer #1: **Yes: **Martin Sparre Andersen

Reviewer #2: No

---

## [Editor Report · Acceptance letter]

17 Dec 2020

PONE-D-20-23700R1 

Relaxation of Social Distancing Restrictions:
Model Estimated Impact on COVID-19 Epidemic in Manitoba, Canada 

Dear Dr. Shafer:

I'm pleased to inform you that your manuscript has been deemed suitable for publication in PLOS ONE. Congratulations! Your manuscript is now with our production department. 

Kind regards, 

on behalf of

Prof. Jeffrey Shaman 

Academic Editor

PLOS ONE